# Protein–protein interaction and non-interaction predictions using gene sequence natural vector

Nan Zhao[1], Maji Zhuo[1], Kun Tian[1] & Xinqi Gong [1,2,3 ✉]

Predicting protein–protein interaction and non-interaction are two important different aspects of multi-body structure predictions, which provide vital information about protein function. Some computational methods have recently been developed to complement experimental methods, but still cannot effectively detect real non-interacting protein pairs. We proposed a gene sequence-based method, named NVDT (Natural Vector combine with Dinucleotide and Triplet nucleotide), for the prediction of interaction and non-interaction. For protein–protein non-interactions (PPNIs), the proposed method obtained accuracies of 86.23% for *Homo sapiens* and 85.34% for *Mus musculus*, and it performed well on three types of non-interaction networks. For protein-protein interactions (PPIs), we obtained accuracies of 99.20, 94.94, 98.56, 95.41, and 94.83% for *Saccharomyces cerevisiae*, *Drosophila melanogaster*, *Helicobacter pylori*, *Homo sapiens,* and *Mus musculus*, respectively. Furthermore, NVDT outperformed established sequence-based methods and demonstrated high prediction results for cross-species interactions. NVDT is expected to be an effective approach for predicting PPIs and PPNIs.

[1] Institute for Mathematical Sciences, School of Mathematics, Renmin University of China, Beijing, China. [2] Beijing Academy of Artificial Intelligence, Beijing, China. [3] Beijing Advanced Innovation Center for Structural Biology, Tsinghua University, Beijing, China. ✉email: xinqigong@ruc.edu.cn

Most vital life activities are related to protein interactions, including physiological and pathological processes[1]. Proteins with similar functions are more likely to interact with each other to form protein complexes that participate in complex and diverse biochemical activities. Detecting and characterizing PPIs provides insight into the functions of unannotated proteins and the mechanisms underlying cellular biochemical processes and complex diseases, and can provide a basis for protein engineering and drug design[2]. The vast majority of proteins do not form interactions in physiological conditions. The number of non-interacting protein pairs vastly exceeds that of interacting pairs. Although some protein pairs have great structural similarities with available experimental complexes, interactions between them can be avoided if they are potentially hazardous to organism[3]. Identifying PPNIs is critical for understanding biological processes and reducing noise in datasets aimed at determining representative features of protein classification. Mastering the biological characteristics of non-interacting protein pairs helps to better study the three-dimensional structure and function of multi-body.

Rapid developments of high-throughput experimental technologies have enabled large-scale discovery and identification of PPIs, but they remain some disadvantages of time-consuming, labor-intensive, and high levels of false-positive and false-negative predictions[4,5]. Therefore, many computational methods have emerged as an alternative for the experimental prediction of PPIs based on different data types, such as genomic information[6], evolutionary information[7,8], structural information[9,10], network information[11], and sequence information[12,13]. Since protein sequences are easily obtained and sequence-based schemes do not need prior knowledge, various sequence-based computational models are favored by researchers. For example, Bock and Gough utilized amino acid physicochemical properties to predict interactions based solely on primary sequences[14]. Shen et al. characterized protein sequence by conjoint triad (CT), where CT considers the properties of adjacent amino acids[15]. Guo et al. expressed protein sequence by auto covariance (AC), which considered interactions between residues separated by a certain distance[16]. Yang et al. employed local descriptors (LD, including composition, transition, and distribution) to evaluate the effect of discontinuous amino acids, but the extracted features ignore global information[17]. Yin et al. numerically represented protein sequences using biochemical properties of amino acids and conducted coevolution analysis based on Fourier transform to detect interacting protein pairs[18]. Due to the difficulty in accurately characterizing protein sequence information by these single feature extraction methods, researchers have proposed many computational methods integrating multiple features for prediction[19,20]. For instance, Zhang et al. used AC, LD, and multi-scale continuous and discontinuous (MCD) local descriptor to extract features information of protein sequence[21]. Chen et al. exploited pseudo amino acid composition, Moreau-Broto, Moran and Geary autocorrelation descriptor, position-specific scoring matrix, and LD to encode biologically relevant features[22]. These feature vectors generally summarize physicochemical properties and position distribution of amino acids, but their large dimensions greatly increase the computational complexity.

The recent advancements in machine learning techniques have been greatly applied in the bioinformatics field, including RNA-binding site identification[23], drugs' efficacy[24], drug–target binding recognition[25], medical diagnosis[26], protein complex structure prediction[27], and protein–protein/peptide/ligand predictions[28–30]. Traditional machine learning algorithms are often combined with feature engineering to detect PPIs, such as amino acid physicochemical properties, co-occurrence frequency[13], CT[15], AC[16,31], LD[17], signature product[32], sequence order, and dipeptide information[33]

were performed with support vector machine (SVM), Gabor feature[34], chaos game representation and wavelet transform[35] were performed with random forest (RF). In addition, deep learning algorithms are also widely used in PPIs detection, such as Siamese-like convolutional neural network (CNN) in DPPI[36] and deep residual recurrent CNN in PIPR[37]. However, deep learning algorithms remain challenging as follows: (i) Lack of interpretability. The deep network is similar to a "black box" in which the physical meaning of features cannot be explained. But traditional machine learning algorithms involve feature engineering, which makes the model easy to interpret and understand. Previous successful methods show that the model generated by the combination of clear sequence features and traditional machine learning performs well. (ii) Complexity and time-consuming. The "inside" of deep learning is difficult to fully understand, which makes hyper-parametric and network design still a considerable challenge. But now that we have a more comprehensive understanding of traditional machine learning underlying algorithms, it is easier to adjust parameters and change model designs. Our lab had effectively captured amino acid sequence information using traditional machine learning algorithms to model and validate PPIs, and had many applications in other biological and medical problems. Yu et al. used SVM with heterogeneous kernels from a specific kernel set, including the Hadamard, RBF, and linear kernels, to perform the breast cancer outcome evaluation and the results proved to be effective[38]. Lyu et al. developed a two-layer SVM ensemble-classifier to predict interface residue pairs of protein trimers and showed its effectiveness and reliability[39]. Wang et al. employed linear SVM, RF, logistic regression with lasso penalty, and logistic regression with hierarchy interaction to predict interface residue pairs respectively, and showed that diverse machine learning methods tend to predict different protein–protein interface patterns[40]. As a result, we select the appropriate feature engineering combined with traditional machine learning algorithms to predict protein–protein interaction and non-interaction.

Plenty of computational models are proposed under the paradigm of supervised learning, so the quality of training data is a key issue to determine the prediction performance. For the training of learning algorithms, interacting protein pairs (i.e., positive samples) and non-interacting protein pairs (i.e., negative samples) are equally important to computational biologists. However, biologists generally focus on interacting protein pairs, extensively collecting experimentally or computationally validated interacting protein pairs into public databases, while ignoring or discarding non-interacting protein pairs[41]. Negative samples in the computational methods are almost constructed by pairing proteins located in different subcellular positions, but these samples restrict the distribution of non-interacting protein pairs and lead to biased estimates of prediction accuracy[42]. Srivastava et al. introduced a triple-layer validation method to collect reliable non-interacting protein pairs and then characterized the most relevant protein correlation features to train the PPIs identification model, which showed excellent predictive capabilities[43]. The training and performance evaluation of models are affected by strong bias in negative samples[44,45], so high-quality and practical negative samples are needed to train less biased models. Smialowski et al. developed the Negatome database, a set of protein pairs unlikely to show direct physical interactions, in 2009 by manually collating the literatures and analyzing three-dimensional protein complex structures[46]. Blohm et al. proposed the second version of this database using a new advanced text-mining process to guide the manual annotation process[47]. High-quality non-interacting samples are important to capture the interaction and non-interaction information from sequences, so Negatome samples are currently used in many studies as an alternative to pairing proteins located in different subcellular. Bryant et al. used negative samples collected in the Negatome database, combined with AlphaFold2, and

optimized multi-sequence alignment to predict heterodimeric protein complexes[48]. Das et al. used a negative dataset from the Negatome database when utilizing interface properties and SVM to classify and differentiate native and non-native complexes[49]. Consequently, the non-interacting protein pairs in the Negatome database can facilitate the training of highly generalizing models to predict PPIs and PPNIs.

Feature selection is a major determinant of the generalizability of predictive models. Common sequence-based computational methods extract features from the amino acid sequence, but nucleotide-related information is missing in the amino acid sequence since multiple codons encode the same amino acid. Triplet nucleotides (i.e., codons) are associated with various diseases, and their repeats may lead to toxic proteins, alter RNA function, and control transcription and translation[50–52]. Moratorio et al.[53] and Carrau et al.[54] found that codon usage may be selected to maintain mutational robustness. Some triple nucleotides often encode the same amino acid, which is called synonymous codons. Codon usage bias, where certain codons are used more frequently than their synonymous codons, is influenced by mutation, selection, and genetic drift. Zhuo et al. found that some interface codons had the obvious propensity to interface residues and the genetic codon affected the interaction interface between proteins[55]. Because codons carry very important information about proteins, some methods consider extracting features from gene sequences. Zhou et al. used the codon pair frequency difference to predict protein interactions with comparable performance to those of other sequence-based methods[56]. Najafabadi et al. utilized the relative codon frequency differences combined with a naive Bayesian classifier to validate PPIs, the approach showed good performance on *Saccharomyces cerevisiae (S. cerevisiae), Escherichia coli*, and *Plasmodium falciparum* datasets[57]. Consequently, extracting additional biological information from gene sequences, like codon frequencies, may further improve the prediction ability. Deng et al. proposed natural vector based on the distributions of nucleotides in each DNA sequence to analyze the virus genome[58]. Later, many methods improved on the natural vector. For example, Dong et al. proposed an improved natural vector method called Accumulated Natural Vector to analyze sequences, genomes, and their phylogenetic relationships[59]. Zhao et al. added the covariances of amino acids to natural vector and used it to classify proteins and detect the evolutionary relationships among species[60]. In addition, certain dinucleotides have connections with the regulation of metabolism, aging, and neurodegeneration[61]. Atkinson et al. provided that dinucleotide bias was related to evading cell defense mechanisms[62]. Takata et al. developed that codon and dinucleotide usage biases may be associated with the need to maintain the RNA secondary structure involved in splicing and gene expression[63]. Kokate et al. studied codon and dinucleotide preferences of 29 *Drosophila* species and observed their association

with speciation[64]. Simón et al. analyzed dinucleotide and codon usage in all available non-redundant viral sequences and discovered that four dinucleotides (TG, GT, CA, and AC) were self-complementary and codon usage bias was mainly determined by the genomic composition[65]. Therefore, we hypothesized that the distribution of contextual nucleotides in a gene sequence could be helpful for sequence classification prediction, so dinucleotide and triplet nucleotide information was added to the natural vector.

In this study, we developed a sequence-based approach for PPIs and PPNIs predictions called NVDT. NVDT firstly employed the distribution of nucleotides, dinucleotides, and triplet nucleotides to extract protein information from gene sequences, where the correspondence between a protein gene sequence and its feature vector was one-to-one. Second, we combined all pairs of corresponding natural vectors into a single feature vector describing a protein pair and then normalized feature vectors using the Z-score method. Finally, these features were fed into the classifier to obtain the final prediction results. NVDT not only combined the advantages of local and global protein information but also had high computational speed with low dimensions, making it a robust and efficient prediction method. We applied our approach to *Homo sapiens (H. sapiens), Mus musculus (M. musculus), S. cerevisiae, Drosophila melanogaster (D. melanogaster),* and *Helicobacter pylori (H. pylori)* datasets, and obtained high prediction results. We harnessed NVDT to produce network visualizations for three types of PPNI networks, including the one-core, multiple-core and crossing network, to further evaluate the capabilities of this approach. In addition, model evaluations indicated that NVDT improved PPIs prediction accuracies over state-of-the-art methods. Our PPNIs prediction results showed that our method was robust and would help to improve multi-body complex structure predictions.

## Results

**Application to *H. sapiens* and *M. musculus* datasets**. To ensure the reliability of our approach, five-fold cross-validation was first used to evaluate model performance and select the optimal parameters (Supplementary Tables 1–4 and Supplementary Figs. 1–4). Then we further verified the performance of different classifiers on the test set. Table 1 showed that the accuracy of the real dataset could be improved substantially when using RF compared with SVM, where the accuracy for the *H. sapiens* increased from 80.92 to 86.23%, and that for *M. musculus* increased from 80.17 to 85.34%. But the differences in accuracy of the constructed dataset were relatively small.

**Network prediction**. We extended our proposed method to predict PPNI networks consisting of non-interaction pairs (NIPs). The knowledge of PPNI networks is helpful to overcome the noise in the datasets and find the relevant features that can

**Table 1 Prediction results by using two classifiers on *H. sapiens* and *M. musculus* datasets.**

| Test set | Classifier | Acc. (%) | Pre. (%) | Sen. (%) | MCC (%) | F-score (%) | AUC |
|---|---|---|---|---|---|---|---|
| *H. sapiens* | | | | | | | |
| real dataset | SVM | 80.92 | 78.83 | 84.54 | 62.00 | 81.59 | 0.8707 |
| | RF | 86.23 | 84.09 | 89.37 | 72.61 | 86.65 | 0.8623 |
| constructed dataset | SVM | 95.41 | 91.59 | 100.00 | 91.21 | 95.61 | 0.9283 |
| | RF | 95.41 | 91.59 | 100.00 | 91.21 | 95.61 | 0.9541 |
| *M. musculus* | | | | | | | |
| real dataset | SVM | 80.17 | 75.36 | 89.66 | 61.46 | 81.89 | 0.8249 |
| | RF | 85.34 | 90.20 | 79.31 | 71.21 | 84.40 | 0.8534 |
| constructed dataset | SVM | 94.83 | 96.43 | 93.10 | 89.71 | 94.74 | 0.9643 |
| | RF | 94.83 | 100.00 | 89.66 | 90.14 | 94.55 | 0.9483 |

best represent proteins, so as to better classify and design the three-dimensional structure of proteins. It can also be used to identify the proteins with the least interaction in a pathway. As a potential indispensable factor in the pathological process, these proteins are likely to be effective targets for drug design. At the same time, systematic analysis of the non-interaction relationship between a large number of proteins in biological systems is very helpful for multi-body complex structure predictions.

We predicted three types of PPNI networks using our approach. First, a one-core network is the simplest network in which only one core protein radially does not interact with other proteins. We found a guanine nucleotide-binding protein, P62879, which may be a modulator or transducer in various transmembrane signaling systems. Among 26 NIPs, 23 pairs were correctly predicted by our method (Fig. 1a and Supplementary Data 1), supporting the application of our method to predict PPNIs in one-core networks.

Second, a multiple-core network is essentially composed of several one-core networks, satisfying the corresponding non-interaction relationships among these nuclear proteins at the same time. We found a PPNI network with the pathway Q8TBX8-O75175-P31150-Q16828-Q8TAU0-Q9H6S3, involving 26 proteins. Of the 82 NIPs in this network, our method correctly predicted 66 pairs (Fig. 1b and Supplementary Data 1). To test whether the lack of core protein non-interaction information leads to low accuracy, we added existing non-interaction information for 10, 30, and 40% of core proteins. Supplementary Table 5 showed the frequency distribution of each core protein that was incorrectly predicted. Using this additional information, the accuracy could be increased from 80.49 to 85.37%, 92.68%, and 96.34%, respectively. With the increase in NIPs related to the six core proteins in the training set, the prediction accuracy of the test set also gradually improved. Therefore, additional experimental information could improve the ability of our method to predict NIPs in complex networks.

Third, in biology, most PPNI networks are crossing networks. We obtained a crossing network involving 73 human proteins from the Negatome database. Our method could correctly predict 58 pairs among 81 NIPs (Fig. 1c and Supplementary Data 1), indicating that our approach can be applied to general PPNI networks.

**Five-fold cross-validation on the constructed dataset**. As there was little difference in accuracy between SVM and RF when using constructed dataset, we used SVM for five-fold cross-validation. As shown in Table 2, the proposed method yielded a high average accuracy of 95.40% for *H. sapien*, 94.83% for *M. musculus*, 98.28% for *S. cerevisiae*, 93.22% for *D. melanogaster*, and 94.63% for *H. pylori*. These findings indicated that our model was effective and robust for PPIs prediction (Supplementary Tables 2, 6 and Supplementary Figs. 2, 5).

**Performance of different feature extraction and combination**. For the same classifier, diverse feature extraction methods may yield different prediction results. To further determine the importance of dinucleotides and triplet nucleotides in predicting protein interactions, we separately predicted interactions combined SVM with NV (using only nucleotide information), NVD (using nucleotide and dinucleotide information), NVT (using nucleotide and triplet nucleotide information), and NVDT (using nucleotide, dinucleotide, and triplet nucleotide information). As shown in Fig. 2, SVM-NVDT showed the best accuracy, precision, MCC, and F-scores for *S. cerevisiae*, *D. melanogaster* and *H. pylori* constructed datasets and high sensitivity (Supplementary Table 7). For *D. melanogaster*, the accuracy of the proposed

method was 7.02% higher than that of SVM-NV, 5.90% higher than that of SVM-NVD, and 1.12% higher than that of SVM-NVT. The polynucleotide module may therefore improve the prediction performance.

Similarly, distinct feature combination methods based on the same classifier will produce different prediction results. For one protein pair composed of protein *i* and protein *j*, the 92-dimensional feature vectors of two proteins can be obtained by the NVDT method. Suppose that the feature vector of protein *i* is $A = (a_1, a_2, …, a_{92})$, and the feature vector of protein *j* is $B = (b_1, b_2, …, b_{92})$. Now we encode the protein pairs in five different means, which are defined as follows.

$$Cod1 : abs(A - B) = [|a_1 - b_1|, |a_2 - b_2|, … , |a_{92} - b_{92}|]$$

$$Cod2 : A + B = [(a_1 + b_1), (a_2 + b_2), … , (a_{92} + b_{92})]$$

$$Cod3 : (abs(A - B), A + B) = [|a_1 - b_1|, … , |a_{92} - b_{92}|, (a_1 + b_1), … , (a_{92} + b_{92})]$$

$$Cod4 : A * B = [(a_1 \times b_1), (a_2 \times b_2), … , (a_{92} \times b_{92})]$$

$$Cod5 : (A, B) = [a_1, a_2, … , a_{92}, b_1, b_2, … , b_{92}]$$

The accuracy results of Cod1–Cod5 on *H. sapiens* and *M. musculus* datasets were shown in Table 3. It can be seen that Cod5 (i.e., our method SVM-NVDT) achieved the highest accuracy in all four datasets.

**Comparison with other feature extraction methods**. The prediction results for alternative methods based on gene sequence data using three datasets were shown in Table 4. Our method showed accuracies of 99.20% for *S. cerevisiae*, 94.94% for *D. melanogaster*, and 98.56% for *H. pylori*, which were better than the two other methods, indicating that NVDT was more suitable for predicting PPIs than other gene sequence-based methods.

We further compared our method with the codon pair-based method(CCPPI) proposed by Zhou et al.[56] on the *S. cerevisiae* dataset. We used the same ten-fold cross-validation as the CCPPI method and obtained an average accuracy of 98.05%, precision of 98.14%, sensitivity of 97.95%, and MCC of 96.10% (Supplementary Table 8). Compared with CCPPI, the average accuracy, precision, sensitivity, and MCC of our method are improved by 8.45, 10.04, 6.25, and 16.80%, respectively.

**Comparison with other sequence-based methods**. In order to make an effective comparison of our proposed SVM-NVDT model, and considering the limited literature based on the *D. melanogaster* dataset, we further compared the predictive performance of our method with other state-of-the-art sequence-based approaches using *S. cerevisiae* and *H. pylori* datasets. The comparison results on the *S. cerevisiae* dataset were presented in Table 5. The accuracy of our proposed method achieved an enhancement of 1.39% compared with the second TAGPPI, 2.11% to the third PIPR, 4.13%to the fourth LightGBM, and 4.56% to the fifth StackPPI. The comparison results on the *H. pylori* dataset were presented in Table 6. Our proposed method achieved an improvement in an accuracy of 7.31% compared with the second PCVMZM, 8.84% to the third StackPPI, 9.09% to the fourth RF+PR+LPQ, and 9.41% to the fifth Weighted Skip-sequential. Our model has shown superior results compared with other methods, further supporting the validity of our model.

**Performance on independent cross-species datasets**. When a large number of interacting proteins in an organism show correlated evolution, orthologs in other taxa will also interact.

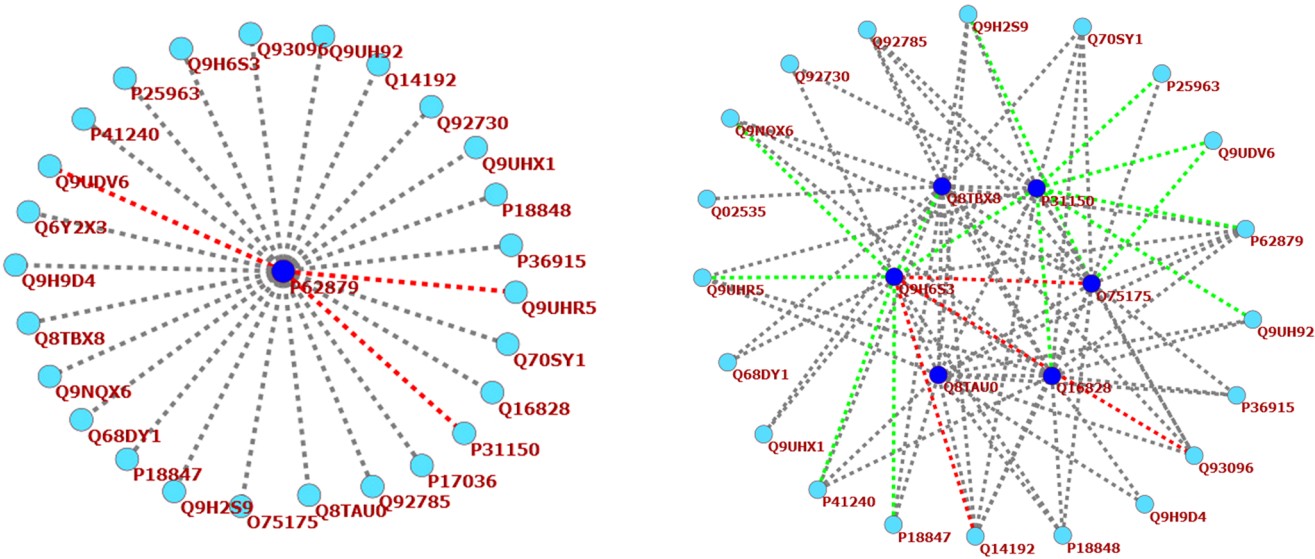

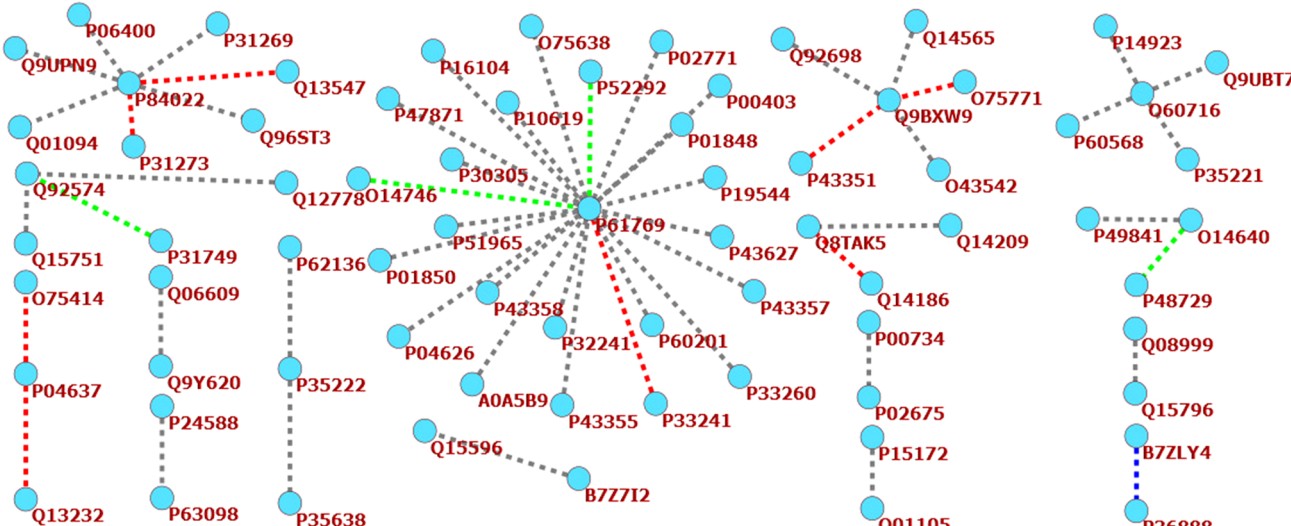

**Fig. 1 PPNI network prediction results. a** A one-core network involving P62879. **b** A multiple-core network involving the Q8TBX8-O75175-P31150-Q16828-Q8TAU0-Q9H6S3 pathway. **c** A crossing network. The core and satellite proteins are represented by indigo blue circles and light blue circles, respectively. Dotted lines connecting two proteins are divided into four classes: gray, predicted correctly; red, predicted falsely; green, re-predicted correctly after adding 40% non-interactions, blue, re-predicted falsely after adding 40% non-interactions.

**Table 2 Five-fold cross-validation results on the constructed dataset.**

| Test set | Acc. (%) | Pre. (%) | Sen. (%) | MCC (%) | F-score (%) | AUC |
|---|---|---|---|---|---|---|
| *H. sapien* | 95.40 ± 1.45 | 91.69 ± 2.39 | 99.90 ± 0.22 | 91.19 ± 2.68 | 95.61 ± 1.33 | 0.9647 ± 0.0360 |
| *M. musculus* | 94.83 ± 2.69 | 96.43 ± 2.64 | 93.10 ± 3.92 | 89.71 ± 5.31 | 94.74 ± 2.77 | 0.9430 ± 0.0184 |
| *S. cerevisiae* | 98.28 ± 0.33 | 98.87 ± 0.47 | 97.68 ± 0.33 | 96.58 ± 0.67 | 98.27 ± 0.33 | 0.9963 ± 0.0009 |
| *D. melanogaster* | 93.22 ± 1.53 | 95.33 ± 1.62 | 90.92 ± 3.03 | 86.57 ± 2.99 | 93.04 ± 1.65 | 0.9656 ± 0.0106 |
| *H. Pylori* | 94.63 ± 2.18 | 97.56 ± 1.64 | 91.56 ± 4.02 | 89.49 ± 4.19 | 94.43 ± 2.33 | 0.9822 ± 0.0088 |

Note: The values in the table are average ± standard deviation.

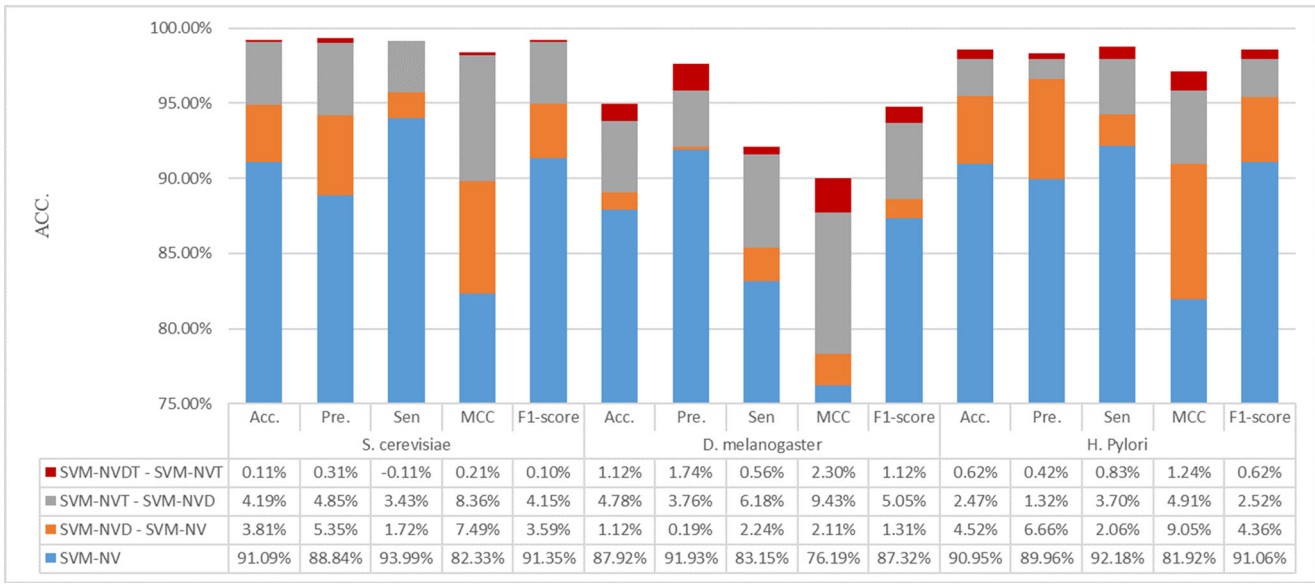

**Fig. 2 Prediction performance based on SVM with NV, NVD, and NVDT on *S. cerevisiae, D. melanogaster* and *H. pylori* constructed datasets.** The abscissa shows the prediction metrics and the ordinate shows the prediction performance.

**Table 3 Prediction performance based on distinct feature combination methods on *H. sapiens* and *M. musculus* datasets.**

| Datasets | Cod1 | Cod2 | Cod3 | Cod4 | Cod5 |
|---|---|---|---|---|---|
| *H. sapiens* constructed dataset | 79.95 | 88.89 | 90.82 | 89.37 | 92.51 |
| *H. sapiens* real dataset | 70.77 | 78.26 | 78.27 | 74.15 | 80.92 |
| *M. musculus* constructed dataset | 86.21 | 89.66 | 90.52 | 88.79 | 94.83 |
| *M. musculus* real dataset | 75.00 | 69.83 | 78.45 | 66.38 | 80.17 |

Note: The values in the table are the accuracy (%) of the independent test set.

Therefore, we trained the model with SVM using all *S. cerevisiae* samples and used the other five species in the DIP database as independent test datasets. For these five test datasets, all samples were interacting protein pairs. As summarized in Table 7, the minimum prediction accuracy was 89.64%, indicating that the proposed method can be used to predict cross-species PPIs.

## Discussion

Our method calibrated on known PPNIs in *H. sapiens* and *M. musculus* addressed the limitations of established computational methods for PPIs prediction, including low accuracies and inefficient prediction of real non-interactions. At the same time, the validity of the model was verified on *S. cerevisiae, D. melanogaster, H. pylori, H. sapiens,* and *M. musculus* constructed datasets based on the commonly used protein pairs using different subcellular locations to construct negative samples. Several aspects of the proposed approach were worth highlighting. (1) We utilized information from gene sequence data, enabling the extraction of more extensive information not accessible in protein sequence data. The dinucleotides and triplet nucleotides are related to many diseases, metabolism, and mutations, and their information can be well obtained by NVDT. (2) NVDT had better performance than those of other gene sequence-based methods, with higher accuracy and better stability and this can be explained by the integration of polynucleotide information, which dramatically reduces the repeatability of different protein feature vectors. (3) NVDT juxtaposed the two feature vectors of a protein pair to

form a new vector with a single dimension, retaining more information. Especially when combined with SVM, the prediction classifications of the two feature vectors (feature_A, feature_B) and (feature_B, feature_A) corresponding to each sample (assuming the protein pair composed of protein A and protein B) are always consistent. The reason may be that each dimension in SVM is independent, and the classification fundamentally depends on calculating the Euclidean distance of any two samples, so the order of features does not affect the classification results. (4) Ensemble classifiers typically show higher accuracy and robust performance than those of single classifiers. However, our model using a single classifier SVM showed better performance than those methods using ensemble classifiers on the *H. pylori* dataset. (5) The constructed negative datasets were restricted to protein pairs located in different cellular compartments, potentially leading to a functional bias in downstream analyses and predictions. The high accuracy of these datasets hardly reflected the true prediction effect, but the good performance on the real dataset can well reflect the feasibility of our model. Among them, results showed that the prediction accuracy of the real dataset was not sufficiently high, which may be explained by the limitation of real non-interacting protein pairs.

The proposed method utilized detailed gene sequence information, including the distribution information of three kinds of nucleotides, and showed better predictive performance than those of established protein sequence-based methods. Moreover, by comparing combined and single features, we found that various features may be complementary. And our model performed well on three types of networks. These results proved that gene sequence information can be used to distinguish interacting and non-interacting protein pairs and ultimately to establish a complete PPI and PPNI map to better understand biochemical and biological processes.

## Methods

In this section, we elaborated on the proposed NVDT approach for predicting PPIs and PPNIs based on gene sequence. NVDT consisted of the following three steps: (1) Encode the protein pairs' interaction and non-interaction information into natural vectors via the distribution of nucleotides, dinucleotides, and triplet nucleotides. (2) Generate and normalize feature vectors of protein pairs. (3) Train the model via different classifiers (SVM and RF) and test on an independent test

**Table 4 Prediction results of different methods on three constructed independent test datasets.**

| Method | Acc. (%) | Pre. (%) | Sen. (%) | MCC (%) | F-score (%) |
|---|---|---|---|---|---|
| *S. cerevisiae* | | | | | |
| Natural vector difference | 78.43 | 83.80 | 70.49 | 57.60 | 76.57 |
| Codon frequency difference | 91.90 | 90.38 | 93.78 | 83.86 | 92.05 |
| our method (SVM-NVDT) | 99.20 | 99.35 | 99.03 | 98.39 | 99.19 |
| *D. melanogaster* | | | | | |
| Natural vector difference | 79.78 | 84.87 | 72.47 | 60.20 | 78.18 |
| Codon frequency difference | 89.04 | 89.71 | 88.20 | 78.10 | 88.95 |
| our method (SVM-NVDT) | 94.94 | 97.62 | 92.13 | 90.03 | 94.80 |
| *H. Pylori* | | | | | |
| Natural vector difference | 71.60 | 71.18 | 71.19 | 43.21 | 71.49 |
| Codon frequency difference | 72.84 | 79.37 | 61.73 | 46.85 | 69.44 |
| our method (SVM-NVDT) | 98.56 | 98.36 | 98.77 | 97.12 | 98.56 |

Note: Natural vector difference method[58]. Codon frequency difference[57].

**Table 5 Comparison of established methods using the *S. cerevisiae* dataset.**

| Model | Acc. (%) | Pre. (%) | Sen. (%) | MCC (%) |
|---|---|---|---|---|
| ACC (Guo, et al., 2008)[16] | 89.33 | 88.87 | 89.93 | N/A |
| AC (Guo, et al., 2008)[16] | 87.36 | 87.82 | 87.30 | N/A |
| Cod1 (Yang, et al., 2010)[17] | 75.08 | 74.75 | 75.81 | N/A |
| Cod2 (Yang, et al., 2010)[17] | 80.04 | 82.17 | 76.77 | N/A |
| Cod3 (Yang, et al., 2010)[17] | 80.41 | 81.66 | 78.14 | N/A |
| Cod4 (Yang, et al., 2010)[17] | 86.15 | 90.24 | 81.03 | N/A |
| SVM+LD (Zhou, et al., 2011)[71] | 88.56 | 89.50 | 87.37 | 77.15 |
| RF+PR+LPQ (Wong, et al., 2015)[72] | 93.80 | 96.66 | 90.64 | 88.35 |
| PCVMZM (Wang, et al., 2017)[73] | 94.48 | 93.92 | 95.13 | 89.58 |
| DeepPPI (Du, et al., 2017)[74] | 94.43 | 96.65 | 92.06 | 88.97 |
| DPPI (Hashemifar, et al., 2018)[36] | 94.55 | 96.68 | 92.24 | N/A |
| LightGBM (Chen, et al., 2019)[20] | 95.07 | 97.82 | 92.21 | 90.30 |
| PIPR (Chen, et al., 2019)[37] | 97.09 | 97.00 | 97.17 | 95.63 |
| StackPPI(Cheng, et al., 2020)[22] | 94.64 | 96.33 | 92.81 | 89.34 |
| TAGPPI(Song, et al., 2022)[75] | 97.81 | 98.10 | 98.26 | 95.63 |
| Our model (SVM-NVDT) | 99.20 | 99.35 | 99.03 | 98.39 |

Note: N/A means not available.

**Table 6 Comparison of existing methods using the *H. pylori* dataset.**

| Model | Acc. (%) | Pre. (%) | Sen. (%) | MCC (%) |
|---|---|---|---|---|
| HKNN (Nanni, 2005)[76] | 84.00 | 84.00 | 86.00 | N/A |
| Signature products (Martin, et al., 2005)[32] | 83.40 | 85.70 | 79.90 | N/A |
| Ensemble of HKNN (Nanni and Lumini, 2006)[77] | 86.60 | 85.00 | 86.70 | N/A |
| Boosting (Shi, et al., 2010)[78] | 79.52 | 81.69 | 80.37 | 70.64 |
| Ensemble ELM (You, et al., 2013)[79] | 87.50 | 86.15 | 88.95 | 78.13 |
| MCD-SVM (You, et al., 2014)[80] | 84.91 | 86.12 | 83.24 | 74.40 |
| Phylogenetic bootstrap (Bock J R et al., 2015)[42] | 75.80 | 80.20 | 69.80 | N/A |
| RF+PR+LPQ (Wong, et al., 2015)[72] | 89.47 | 89.63 | 89.18 | 81.16 |
| PCVMZM (Wang, et al., 2017)[73] | 91.25 | 90.06 | 92.05 | 84.04 |
| DeepPPI (Du, et al., 2017)[74] | 86.23 | 84.32 | 89.44 | 72.63 |
| Weighted Skip-sequential (Goktepe and Kodaz, 2018)[81] | 89.15 | 87.29 | 88.13 | 77.21 |
| LightGBM (Chen, et al., 2019)[20] | 89.03 | 88.36 | 89.99 | 78.14 |
| StackPPI (Cheng, et al., 2020)[22] | 89.72 | 90.37 | 87.93 | 78.59 |
| Our model (SVM-NVDT) | 98.56 | 98.36 | 98.77 | 97.12 |

Note: N/A means not available.

set. Finally, we briefly described the common model performance evaluation metrics used in the study. The flowchart of NVDT was shown in Fig. 3.

**Dataset collection**. Seven datasets were obtained. The positive datasets consisted of interacting protein pairs collected from the public Database of Interacting Proteins (DIPs: https://dip.doe-mbi.ucla.edu/dip/)[66]. To reduce fragments and sequence similarity, samples with fewer than 50 amino acids and >40% pairwise sequence identity to one another were excluded.

The negative datasets were composed of non-interacting protein pairs obtained in two ways. First, negative samples were derived from the Negatome Database 2.0 (http://mips.helmholtz-muenchen.de/proj/ppi/negatome)[47], which currently

contained experimentally supported non-interacting protein pairs. In the Negatome database, the subcellular structure information for some proteins was unclear or even absent, and some proteins belonged to at least two or more subcellular localizations simultaneously, which was consistent with the actual phenomenon. After a selection of protein pairs from multiple species in the Negatome database, the majority belonged to *H. sapiens* (1217 pairs), followed by *M. musculus* (347 pairs) and *Rattus norvegicus* (33 pairs). Accordingly, we collected 2434 protein pairs for *H. sapiens* and 694 protein pairs for *M. musculus*, with interacting protein pairs and non-interacting protein pairs each accounting for half. The datasets obtained in this way were called "real dataset".

The other negative samples in different subcellular compartments were obtained, based on the assumption that proteins within different subcellular localizations tend not to interact. Considering that the ratio of positive samples to negative samples used in previous literature research is mostly 1:1, we should not only verify the prediction accuracy of interacting protein pairs but also ensure the prediction accuracy of non-interacting protein pairs. Therefore, the balanced dataset was selected when constructing the dataset, that is, we randomly selected negative samples with the same number of positive samples. Here, the selection of negative samples was random without any cluster analysis. Four datasets were finally collected in this way, including 11188 protein pairs for *S. cerevisiae*, 2140 protein pairs for *D. melanogaster*, 1217 protein pairs for *H. sapiens*, and 694 protein pairs for *M. musculus*. We also performed on 2916 protein pairs for *H. pylori* described derived by Rain et al.[67], and a list of *H. pylori* protein interactions was given in its supplementary materials. These datasets were called "constructed dataset".

The gene sequence of each protein was obtained from the NCBI database (https://www.ncbi.nlm.nih.gov/). To verify the performance of the proposed method, the positive samples of *H. sapiens* and *M. musculus* datasets in the real dataset and constructed dataset were consistent, only the negative samples were obtained differently (Supplementary Table 9). Finally, each of the seven datasets was divided into a training dataset and a test dataset in a ratio of 5:1.

**Table 7 Prediction results for independent datasets.**

| Test species | No. of Test pairs | Acc. (%) |
|---|---|---|
| *D. melanogaster* | 1070 | 99.35 |
| *H. pylori* | 1458 | 94.51 |
| *Caenorhabditis elegans* | 1045 | 98.56 |
| *M. musculus* | 336 | 98.50 |
| *Escherichia coli* | 2143 | 89.64 |

**Feature extraction.** Deng et al. have defined natural vector and mathematically proved that there was a one-to-one correspondence between natural vector and gene sequence for a protein[58]. The natural vector method to extract vital information from gene sequence was as follows.

Let $S = s_1 s_2 \cdots s_N$ be a gene sequence of length $N$, where $s_i \in \{A, C, G, T\}, i = 1, 2, \ldots, N$. For $k$ representing one of the four nucleotides, define $\omega(\cdot) = \{A, C, G, T\} \rightarrow \{0, 1\}$, where

$$\omega_k(s_i) = \begin{cases} 0, s_i \neq k, \\ 1, s_i = k. \end{cases} \quad (1)$$

(1) Denote $n_k = \sum_{i=1}^N \omega_k(s_i)$ as the number of nucleotide $k$ in gene sequence $S$.
(2) Taking the first nucleotide as the origin, $\text{dis}_{k(i)} = i \cdot \omega_k(s_i)$ is the distance from the first nucleotide to the $i$th nucleotide $k$, where $i = 1, 2, \ldots, n_k$, and $\mu_k = \frac{\sum_{i=1}^{n_k} \text{dis}_{k(i)}}{n_k}$ represents the average position of nucleotide $k$ in gene sequence $S$.
(3) The normalized central moments are defined as $D_j^k = \sum_{i=1}^{n_k} \frac{(i - \mu_k)^j \cdot \omega_k(s_i)}{n_k^{j-1} N^{j-1}}, j = 1, 2, \ldots, n_k$.

The first normalized central moments $D_1^k$ ($j = 1$ in $D_j^k$) could be ignored since its values were zero. Deng et al. and Yu et al. have demonstrated that the second normalized central moments $D_2^k$ ($j = 2$ in $D_j^k$) in the vector could obtain stable classification results; accordingly, the central moments ($D_j^k$) higher than $j = 2$ do not need to be considered, and computed[58,68]. Thus, the 12-dimensional natural vector $N(S)$ of a gene sequence $S$ was given as follows:

$$N(S) = \left(n_A, n_C, n_G, n_T, \mu_A, \mu_C, \mu_G, \mu_T, D_2^A, D_2^C, D_2^G, D_2^T\right) \quad (2)$$

In this work, we proposed an extended natural vector with the frequencies of dinucleotides and triplet nucleotides, which considered the properties of a single nucleotide and its vicinal nucleotides and regarded any two contiguous nucleotides or any three contiguous nucleotides as a unit. The size of type dinucleotide should be $4 \times 4 = 16$ and the size of type triplet nucleotide should be $4 \times 4 \times 4 = 64$. Finally, we defined a 92-dimensional natural vector to make further investigation on gene sequence as the following:

$$N(S) = \left(n_A, n_C, n_G, n_T, \mu_A, \mu_C, \mu_G, \mu_T, D_2^A, D_2^C, D_2^G, D_2^T, n_{AA}, \ldots, n_{TT}, n_{AAA}, \ldots, n_{TTT}\right) \quad (3)$$

For each protein pair in the dataset, we could convert a gene sequence into a natural vector. Assume that the $k$th protein pair corresponds to protein $i$ and protein $j$, we then juxtaposed the two natural vectors. The features of the protein

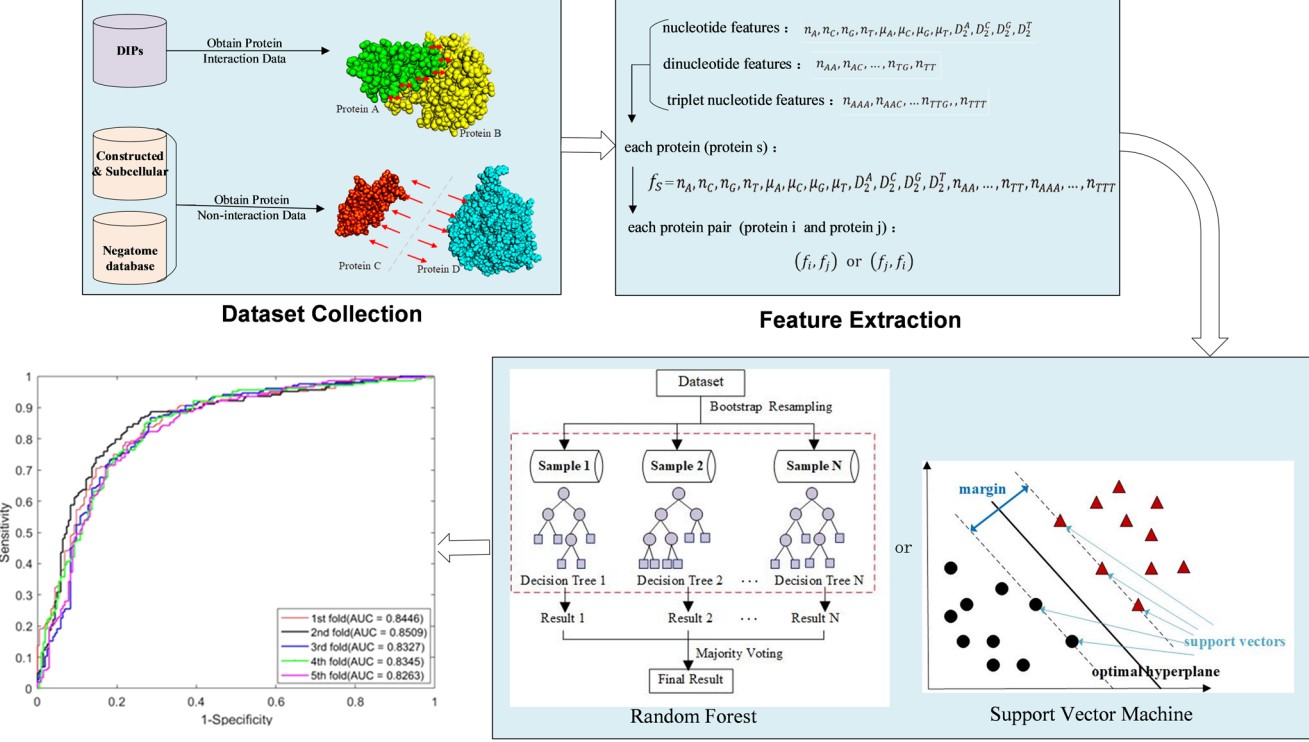

**Fig. 3 Workflow of our computational pipeline to predict protein-protein interaction and non-interaction.** It describes the whole research process, including dataset collection, feature extraction, feature-selective classifier and result analysis.

pair can be described as two 184-dimensional feature vectors:

$$d_k = (N_i, N_j) \text{ or } (N_j, N_i) \quad (4)$$

where $N_i$ and $N_j$ were feature vectors for proteins $i$ and $j$.

However, protein feature vectors correlated to the length of protein (the number of nucleotides) complicate the comparison between two protein pairs. The feature vectors were standardized by the Z-score method[69], with the mean value of 0 and standard deviation of 1:

$$d_k'(l) = \frac{d_k(l) - \mu(l)}{\sigma(l)}, k = 1, 2, \dots, n, \text{ and } l = 1, 2, \dots, 184 \quad (5)$$

where $d_k(l)$ was the $l$th feature of the $k$th protein pair, and $n$ was the number of protein pairs. $\mu(l)$ and $\sigma(l)$ were the mean value and standard deviation of all proteins for the $l$th feature, respectively:

$$\mu(l) = \frac{\sum_{k=1}^{n} d_k(l)}{n}, \sigma(l) = \sqrt{\frac{\sum_{k=1}^{n} \left(d_k(l) - \mu(l)\right)^2}{n}} \quad (6)$$

For data standardization, the training set was standardized in the above way, and the mean and standard deviation of the training set were used to standardize the test set. $d_k'$, the input features of a classifier, were a 184-dimensional vector containing the statistical features of nucleotide, dinucleotide, and triplet nucleotide.

**Support vector machine**. SVM[70] is a supervised learning method that has been popularly utilized for classification and regression in computational biology. It constructs a hyperplane to maximize the margin. Specifically, SVM finds several samples as support vectors that minimize the distance of samples between different classes, and the minimum distance is called the margin. The optimal hyperplane is located in the center of the margin, where the larger the margin is, the smaller the generalization error will be. Therefore, this hyperplane can better divide the samples in the training set according to the labels given. In this study, the Gaussian radial basis function was chosen as the kernel function, and the regularization parameter c and the Gauss kernel function parameter γ were optimized by a grid search approach. Here, the parameters were the default of LIBSVM, and cross-validation was used to avoid overfitting (Supplementary Table 10). And select a cut-off value of 0.5. The application of the SVM classifier under optimal parameter settings guarantees the reliability of classification prediction with the minimum error.

**Random forest**. Random forest (RF)[70] is another typical classification and regression method with biological applications. It uses multiple decision trees, which do not correlate them, to obtain the final result by voting or taking the mean value, generating an overall model with high accuracy and generalizability. For a new sample, each decision tree assigns the sample to a class, and then the voting method is used to determine which class is selected more frequently as the final classification result. Three parameters are usually adjusted when training an RF model, the number of trees to grow (denoted as n_Tree), the number of randomly selected features at each decision split (denoted as n_feature), and the minimum node size of a terminal node (denoted as min_leaf). In our study, n_Tree values under 200 were evaluated to find the optimal value concerning computational time, cost, and overfitting (Supplementary Table 10). All other parameters were set to default values.

**Performance evaluation**. Common performance evaluation metrics were used, including accuracy (Acc.), precision (Pre.), sensitivity (Sen.), specificity (Spe.), Matthews correlation coefficient (MCC), F1-score, and the area under a ROC curve (AUC) to evaluate the predictive performance of the proposed method. These metrics were defined as follows:

$$\text{Accuracy} = \frac{TP + TN}{TP + TN + FP + FN} \quad (7)$$

$$\text{Precision} = \frac{TP}{TP + FP} \quad (8)$$

$$\text{Sensitivity} = \frac{TP}{TP + FN} \quad (9)$$

$$\text{Specificity} = \frac{TN}{TN + FP} \quad (10)$$

$$\text{MCC} = \frac{TP \times TN - FP \times FN}{\sqrt{(TP + FN) \times (TN + FP) \times (TP + FP) \times (TN + FN)}} \quad (11)$$

$$\text{F1-score} = \frac{2 * \text{precision} * \text{sensitivity}}{\text{precision} + \text{sensitivity}} \quad (12)$$

where $TP$ represents the number of positive samples that are correctly predicted; $FN$ represents the number of positive samples that are incorrectly predicted; $TN$ represents the number of negative samples that are correctly predicted; $FP$ represents the number of negative samples that are incorrectly predicted. The receiver operating characteristic (ROC) curve was generated by plotting the $TP$ rate against the $FP$ rate at various thresholds; the abscissa of the ROC curve was 1-specificity and the longitudinal coordinate is sensitivity.

**Statistics and reproducibility**. All quantitative results of K-fold cross-validation were shown using mean ± standard deviation. Graphing of three types of PPNI networks was performed with Pajek. The computational experiments were repeated at least two additional times with a similar outcome.

**Reporting summary**. Further information on research design is available in the Nature Research Reporting Summary linked to this article.

## Data availability

The data used in this work are available at https://github.com/Zhaonan99/NVDT and Supplementary Data 1. Supplementary Data 1 is a .xlsx file that includes data for reproducing Fig. 1. Supplementary Table 7 includes data for reproducing Fig. 2. The interacting protein pairs discussed have been deposited in the DIP database, the real non-interacting protein pairs are accessible through the Negatome Database 2.0, and the gene sequence of each protein is available from the NCBI database. Other information is available from the corresponding author on reasonable request.

## Code availability

The source code is available at https://github.com/Zhaonan99/NVDT.

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

## Acknowledgements

This work was supported by the National Natural Science Foundation of China (No. 31670725), the Beijing Advanced Innovation Center for Structural Biology of Tsinghua University, and the Public Computing Cloud of Renmin University of China.

## Author contributions

N.Z. developed the method, collected the data, did the computation, and wrote the initial manuscript and later revised manuscripts. M.J.Z. proposed to draw on the natural vector and collected *Saccharomyces cerevisiae* dataset. K.T. helped to revise the manuscript. X.Q.G. designed the project and revised the manuscript. All authors read and approved the final manuscript.

## Competing interests

The authors declare no competing interests.
