## [Peer Review File · Communications Biology]

Reviewers' comments:

Reviewer #1 (Remarks to the Author):

The author proposed a new method NVDT to predict the protein-protein interaction sites from gene sequences. The new method used the composition of nucleotides, di-nucleotides, and triplet nucleotides, nature vectors of protein pairs. The new method obtained high prediction results on Homo sapiens, Mus musculus, Saccharomyces cerevisiae, Drosophila melanogaster, and Helicobacter pylori datasets.

I have following questions,

1. There are two types of datasets: "constructed dataset", "real dataset" which is the training dataset and which is the test datasets.
2. In Section 3.1 "To ensure the reliability of our approach, five-fold cross-validation was first used to select the optimal parameters. Then we further verified the performance of different classifiers on the test set. " Do you optimize parameters on the test set? Which is the training set ?
3. In table 2, SVM obtained AUC value of 0.8707, RF got AUC value of 0.8672 on the real dataset of H. sapiens. Why did you write that "RF classifier performed better than the SVM classifier on the real datasets"?
4. Five-fold cross validation results were shown in table 4. However, ten-fold cross validation results were shown in table 6. Why not use the same fold number? In fact, it is hard to show your method SVM-NVDT is better than CCPPI with 10-fold cross validation since the dataset was split into ten folds very differently. You can try to use to compare SVM-NVDT and CCPPI with the Jackknife test in Table 6.
5. The new method was not well evaluated. In Table 7, TAGPPI, a quite new method published in 2022, was compared with SVM-NVDT on S. cerevisiae dataset. However, TAGPPI was not listed in Table 8 for H. pylori dataset.
6. In Line 209, Line 213. "Eigenvectors" is not correct. You did not compute the eigenvalues, and eigenvectors. It could be changed "feature vector"
7. The Introduction of the paper is poor. Lots of new published methods should be discussion.

Reviewer #2 (Remarks to the Author):

PPIs have been extensively studied. Many predictive models have been developed (see <https://www.nature.com/articles/s42256-020-0149-6.pdf>). The authors consider the protein-protein non-interaction (PPNI) problem in this work. They developed a novel sequence-based approach for protein-protein interaction and non-interaction predictions called NVDT. The results are quite interesting. I have some minor concerns that should be addressed.

- 1) The definitions of accuracy (Acc.), precision (Pre.), sensitivity (Sen.), specificity (Spe.), Matthews correlation coefficient (MCC), and F-score are trivial and should be included only in the supporting information.
- 2) Stephen Yau and co-authors have defined natural vectors. The similarity and difference should be discussed.
- 3) SVM and RF are very simple methods. Why did not the authors try more sophisticated methods like deep learning for the prediction?
- 4) The datasets used in this work should be summarized in a Table, including their properties, like sequence maximal and minimal lengths for each dataset, number of protein pairs, etc.
- 5) Kmers-based methods are often used in this subject. The authors may compare the performance of their approach to that of k-mers based methods.

Reviewer #3 (Remarks to the Author):

The paper presents a novel machine learning prediction for protein-protein interaction (PPI) using the gene sequences only. The method may effectively predict PPIs and non-PPIs, and shows promising applications in protein function research. I would have following concerns that the paper shall address.

- Protein-protein interactions depend directly on protein sequences, not directly on gene sequences. Why the feature builds are on the gene sequences, not on protein sequences? The paper shall answer this question, at least show that the gene sequences have same performance on protein sequences. In addition, the paper concludes that di or tri- nucleotides are important for PPI prediction, what are the connection and impacts of the di- and tri-mers in actual protein sequences? It seems a gap between the nucleotide distributions and protein sequences.
- Protein-protein interaction is two directions, that means when protein A interacts with B, equally, protein B interacts with A. In feature extraction of PPI, the paper juxtaposes and concatenates two vectors A and B as either [A, B] or [B, A]. This concatenated feature is ambiguous because we do not know which one is determinant. It is a problem in actual prediction using this concatenated feature, for example, when two interaction-unknown proteins C and D, which feature [C, D] or [D,C] can be used in prediction using the models? This is critical problem that paper shall address carefully. One possible solution is to add or element-wise product of two feature vectors from two proteins, not concatenate the features.
- When constructing features of PPI, the paper uses Z-score of the features, but the Z-score uses l-th eigenvalue of the protein pair, why l-th eigenvalue is used? Where the eigenvalue is from? is it from the eig-decomposition of the covariance matrix of the feature? Please justify the formula definition.
- Two ML methods SVM and RF are used in the paper, but I am not sure what is the purpose of SVM in the method. Does the paper combines the two methods by selecting the best one? or just try to explore which ML method is better?
- The datasets for negative PPI are from authors' collection and public curated dataset (Negatome), are the manually collected PPI dataset validated by literature searching (or experiments)? If not, the manually collected PPI dataset have more noise. The quality of dataset is very important for ML models.

ANSWER TO REVIEWERS COMMENTS

We would like to thank the reviewers for providing us with many fascinating discussion points, which enabled us to considerably improve our manuscript. Here, we provide a detailed answer to all comments. The following four aspects are our major revisions:

1. Five-fold cross-validation performance evaluation results for each species on the training set in the Results part and the Supplementary Material part.
2. Parameter selection used Support Vector Machines (SVM) and Random Forests (RF) for each species on the training set in the Materials and methods part and the Supplementary Material part.
3. Recent research progress on protein-protein interactions and non-interactions prediction methods, and research progress on traditional machine learning/deep learning, negative samples selection, feature extraction, as well as the related contents of natural vector, dinucleotide and triplet nucleotide in the Introduction part.
4. Correct errors in presentation and adjust the order.
5. Comparison of experimental results and datasets information supplement.

Reviewers' comments

Reviewer #1 (Remarks to the Author):

The author proposed a new method NVDT to predict the protein-protein interaction sites from gene sequences. The new method used the composition of nucleotides, di-nucleotides, and triplet nucleotides, nature vectors of protein pairs. The new method obtained high prediction results on *Homo sapiens*, *Mus musculus*, *Saccharomyces cerevisiae*, *Drosophila melanogaster*, and *Helicobacter pylori* datasets.

I have following questions,

1. There are two types of datasets:, “constructed dataset“, “real dataset” which is the training dataset and which is the test datasets.

Thank you for the question. The constructed dataset contains datasets for five species, while the real dataset contains datasets for two species. Therefore, we finally obtained 7 datasets for training model and testing respectively, including *H. sapiens* constructed dataset, *M. musculus* constructed dataset, *S. cerevisiae* constructed dataset, *D. melanogaster* constructed dataset, *H. pylori* constructed dataset, *H. sapiens* real dataset and *M. musculus* real dataset. Each of the seven datasets was divided into a training dataset and a test dataset according to the ratio of 5:1.

Take *D. melanogaster* constructed dataset as an example, 2140 samples (1070 positive samples and 1070 negative samples) were divided into training set and test set according to the ratio of 5:1, while maintaining the same number of positive and negative samples. Finally,

we got the training set (892 positive samples and 892 negative samples) and the test set (178 positive samples and 178 negative samples).

2. In Section 3.1 “To ensure the reliability of our approach, five-fold cross-validation was first used to select the optimal parameters. Then we further verified the performance of different classifiers on the test set. “ Do you optimize parameters on the test set? Which is the training set ?

(1) We did not optimize parameters on the test set. The parameters selection depends on the results of five-fold cross-validation on the training set, which are not related to the test set.

We used different parameters to conduct multiple five-fold cross-validation on the training set. The result of each five-fold cross-validation was taken from the average of the accuracy of the five validation sets. Then the results of multiple five-fold cross-validation were sorted and the parameters corresponding to the high-precision results were finally selected. Next, we used these parameters to train the model on the whole training set, and test it on an independent test set.

(2) Each dataset of seven datasets was divided into a training dataset and a test dataset in a ratio of 5:1.

Take *D. melanogaster* constructed dataset as an example, which contains 2,140 samples with 1,070 positive samples and 1,070 negative samples. When we divided the whole dataset into a training set and a test set according to the ratio of 5:1, we should ensure that the positive samples and negative samples in each dataset have the same equality. Therefore, we randomly select four fifths of the 1,070 positive samples as the positive samples of the training set, and four fifths of the 1,070 negative samples as the negative samples of the training set. In this way, we finally got the training set, which contained 892 positive samples and 892 negative samples.

3. In table 2, SVM obtained AUC value of 0.8707, RF got AUC value of 0.8672 on the real dataset of *H. sapiens*. Why did you write that “RF classifier performed better than the SVM classifier on the real datasets”?

Thank you for the suggestion. In the work, we have evaluated the models performance on seven predictors, including accuracy (Acc.), precision (Pre.), sensitivity (Sen.), specificity (Spe.), Matthews correlation coefficient (MCC), F1-score and the area under an ROC curve (AUC). In the first six indicators, RF is all better than SVM, and only in AUC, the effect of RF is 0.0035 lower than SVM. In addition, ACC and MCC, which we pay most attention to, were superior to SVM on RF. Therefore, we simply generalized that RF classifier performs better than SVM classifier on the real datasets.

4. Five-fold cross validation results were shown in table 4. However, ten-fold cross validation results were shown in table 6. Why not use the same fold number? In fact, it is hard to show your method SVM-NVDT is better than CCPPI with 10-fold cross validation since the dataset was split into ten folds very differently. You can try to use to compare SVM-NVDT and CCPPI with the Jackknife test in Table 6.

Thank you for the suggestion.

(1) K-fold cross validation is often used to evaluate the proposed model performance and select model parameters. Most previous literatures adopted the 5-fold cross validation, because it was not as time-consuming and laborious as 10-fold cross validation, and was sufficient to evaluate model performance and select parameters. Therefore, in this work, we adopted 5-fold cross validation in all datasets (including table 4). Since CCPPI experimental literature adopted 10-fold cross validation, to compare with its results, we also conducted ten-fold cross validation on *Saccharomyces cerevisiae* dataset (table 6).

(2) The 10-fold cross validation divides the training set into 10 parts on average, each consisting of equal number of positive samples and negative samples. Any 9 parts are used for training and the other part is used for validation. Finally, the results of 10 models on their respective validation sets are averaged to obtain the 10-fold cross validation performance. The results of the 10-fold cross validation can well evaluate the generalization ability of the model.

5. The new method was not well evaluated. In Table 7, TAGPPI, a quite new method published in 2022, was compared with SVM-NVDT on *S. cerevisiae* dataset. However, TAGPPI was not listed in Table 8 for *H. pylori* dataset.

Thank you for the suggestion. The results of SVM-NVDT method were compared with the TAGPPI method on the *S. cerevisiae* dataset because TAGPPI performed on the same size *S. cerevisiae* dataset and published the testing results. We did not compare the SVM-NVDT method with the TAGPPI method on the *H. pylori* dataset because the TAGPPI was not tested on the *H. pylori* dataset. But we compared SVM-NVDT method with a variety of other computational methods on the *H. pylori* dataset.

6. In Line 209, Line 213. "Eigenvectors" is not correct. You did not compute the eigenvalues, and eigenvectors. It could be changed "feature vector"

We thank the reviewer for pointing out this error, which is corrected in the revised manuscript. We have changed "eigenvectors" to "feature vector".

7. The Introduction of the paper is poor. Lots of new published methods should be discussion.

Thank you for the suggestion. We have revised the introduction. We have added new published literatures, and analyzed and summarized different methods from dataset, features and algorithms aspects, and added. Furthermore, we added the origin and application of natural vector, as well as the reasons why we chose the method based on natural vector to extract sequence features.

Reviewer #2 (Remarks to the Author):

PPIs have been extensively studied. Many predictive models have been developed (see <https://www.nature.com/articles/s42256-020-0149-6.pdf>). The authors consider the protein-protein non-interaction (PPNI) problem in this work. They developed a novel sequence-based approach for protein-protein interaction and non-interaction predictions called NVDT. The results are quite interesting. I have some minor concerns that should be addressed.

1) The definitions of accuracy (Acc.), precision (Pre.), sensitivity (Sen.), specificity (Spe.), Matthews correlation coefficient (MCC), and F-score are trivial and should be included only in the supporting information.

Thank you for the suggestion. We have moved the detailed definitions of these statistics measures to the supporting materials.

2) Stephen Yau and co-authors have defined natural vectors. The similarity and difference should be discussed.

Thank you for the suggestion. In the introduction and methods section, we have supplemented and discussed similarity and difference of natural vector defined by Stephen Yau et al. and NVDT proposed in this paper.

3) SVM and RF are very simple methods. Why did not the authors try more sophisticated methods like deep learning for the prediction?

Traditional machine learning algorithms, such as SVM and RF, have benefits as follows:

(i) Interpretability. The deep network is similar to a "black box" in which the physical meaning of the features it learns cannot be explained. However, traditional machine learning algorithms involve feature engineering, which makes the model easy to interpret and understand. Previous successful methods show that models with clear sequence features have good performance when combined with traditional machine learning.

(ii) Simplicity and time-saving. The "inside" of deep learning is difficult to fully understand, which makes hyper-parametric and network design still a considerable challenge. But now that we have a more comprehensive understanding of the machine learning underlying algorithms, it is easier to adjust parameters and change model designs.

In this work, we have clear protein sequence features based on natural vector, so combined with traditional machine learning is a good choice.

4) The datasets used in this work should be summarized in a Table, including their properties, like sequence maximal and minimal lengths for each dataset, number of protein pairs, etc.

Thank you for the suggestion. We have added sequence maximal and minimal lengths for each dataset to the Table 1. At present, Table 1 gives the following information of each dataset: the species of proteins, the number of interacting protein pairs, the number of non-interacting protein pairs, sequence maximal and minimal lengths for each dataset, and the source of protein pairs collected.

5) Kmers-based methods are often used in this subject. The authors may compare the performance of their approach to that of k-mers based methods.

Thank you for the suggestion. We carefully study and use kmers-based methods.

(1) Many kmers-based methods have been applied in biological evolutionary analysis, which mainly analyze the frequency of k-letter strings in the protein sequence [1]. Feature Frequency Profiles (FFP) is a classical method, in which the selection of k is very important [2]. The FFP method gives the selection method of k value: select k as the minimum integer value of $\log_4 N$, that is,

$$k = \text{floor}(\log_4 N),$$

where N is the maximum length of the sequence in the dataset studied [2].

Now consider the length of the sequence in the datasets studied in this work. See the following table for details.

Dataset Name	Dataset Name	Sequence maximal length	Sequence minimal length
real dataset	H. sapiens	568,080	510
	M. musculus	1,125,620	315
constructed dataset	S. cerevisiae	14,733	168
	D. melanogaster	394,146	195
	H. pylori	9,562	183
	H. sapiens	453,160	510
	M. musculus	1,125,620	725

The sequence maximal length obtained on the M. musculus dataset was 1125620. According to the above method, the value of k can be calculated, $k = \text{floor}(\log_4 1125620) \approx 9$. And the number of string combinations of k can be calculated, $4^9 = 262144$. That is, the dimension of each protein feature vector is 262144. Finally, the dimension of each protein pair feature vector is $262144 \times 2 = 524288$.

Even considering the minimum value 9562 of sequence length in the H. pylori dataset, the value of k can be calculated, $k = \text{floor}(\log_4 9562) \approx 6$. And the number of string combinations of k can be calculated, $4^6 = 4096$. That is, the dimension of each protein

feature vector is 4096. Finally, the dimension of each protein pair feature vector is $4096 \times 2 = 8192$.

The dimension of the feature vectors obtained by this method is particularly large, which will increase the computation complexity.

(2) The nucleotide, dinucleotide and triplet nucleotide in this paper correspond to the cases when $k = 1, 2, 3$ in the kmers-based method. In section 3.4, the results of different feature extraction were compared. It can be inferred that the results obtained by using single nucleotide frequency, or dinucleotide frequency, or triplet nucleotide frequency alone are not as good as their combination. At the same time, the accuracy of combining features on *S. cerevisiae*, *D. melanogaster* and *H. pylori* datasets can reach 99.20%, 94.94% and 98.56%.

Reference:

- [1] Blaisdell BE. Average values of a dissimilarity measure not requiring sequence alignment are twice the averages of conventional mismatch counts requiring sequence alignment for a computer-generated model system. *J Mol Evol.* 1989 Dec;29(6):538-47.
- [2] Sims GE, Jun SR, Wu GA, Kim SH. Alignment-free genome comparison with feature frequency profiles (FFP) and optimal resolutions. *Proc Natl Acad Sci U S A.* 2009 Feb 24;106(8):2677-82.

Reviewer #3 (Remarks to the Author):

The paper presents a novel machine learning prediction for protein-protein interaction (PPI) using the gene sequences only. The method may effectively predict PPIs and non-PPIs, and shows promising applications in protein function research. I would have following concerns that the paper shall address.

- Protein-protein interactions depend directly on protein sequences, not directly on gene sequences. Why the feature builds are on the gene sequences, not on protein sequences? The paper shall answer this question, at least show that the gene sequences have same performance on protein sequences. In addition, the paper concludes that di or tri- nucleotides are important for PPI prediction, what are the connection and impacts of the di-and tri-mers in actual protein sequences? It seems a gap between the nucleotide distributions and protein sequences.

Thank you for the suggestion. Gong et al. found that some interface codons have obvious propensity to interface residues, which demonstrated that the genetic codon does affect interaction interface between proteins. Furthermore, two or three consecutive nucleotides occur at different frequencies in different genomes, and triplet nucleotides are associated with various diseases. For these reasons, we capture protein features based on gene sequences. Moreover, results on *S. cerevisiae* dataset and *H. pylori* constructed dataset show that the performance of our gene sequence-based model is comparable to that of the amino acid sequence-based model. We have made a detailed supplement in the introduction section.

- Protein-protein interaction is two directions, that means when protein A interacts with B, equally, protein B interacts with A. In feature extraction of PPI, the paper juxtaposes and concatenates two vectors A and B as either [A, B] or [B, A]. This concatenated feature is ambiguous because we do not know which one is determinant. It is a problem in actual prediction using this concatenated feature, for example, when two interaction-unknown proteins C and D, which feature [C, D] or [D,C] can be used in prediction using the models? This is critical problem that paper shall address carefully. One possible solution is to add or element-wise product of two feature vectors from two proteins, not concatenate the features.

Thank you for the suggestion. There are two main reasons for selecting concatenated features:

(i) Feature independence.

The features extracted from protein gene sequences are independent of each other. Therefore, the feature vector of protein pairs is not affected by the positions of the feature vectors of the corresponding two proteins. And many previous studies have also used concatenated features [1-3].

In addition, we extracted two concatenated feature vectors, (A, B) and (B, A), of the protein pairs from *H. sapiens*, *M. musculus*, *S. cerevisiae*, *D. melanogaster*, and *H. pylori*

datasets. In the independent test set, the labels predicted by the two concatenated feature vectors of the same protein pair remained consistent.

(ii) Comparative test.

We also compared the different combinations of protein feature vector combined with SVM on *H. sapiens* and *M. musculus* datasets. Suppose we have two proteins: protein A and protein B, where $A = (a_1, a_2, \dots, a_{92}), B = (b_1, b_2, \dots, b_{92})$. Now we encode the protein pairs in five different means, which are defined as follows.

Cod1: $\text{abs}(A - B) = [|a_1 - b_1|, |a_2 - b_2|, \dots, |a_{92} - b_{92}|]$

Cod2: $A + B = [(a_1 + b_1), (a_2 + b_2), \dots, (a_{92} + b_{92})]$

Cod3: $[\text{abs}(A - B), A + B] = [|a_1 - b_1|, \dots, |a_{92} - b_{92}|, (a_1 + b_1), \dots, (a_{92} + b_{92})]$

Cod4: $A * B = [(a_1 \times b_1), (a_2 \times b_2), \dots, (a_{92} \times b_{92})]$

Cod5: $(A, B) = [a_1, a_2, \dots, a_{92}, b_1, b_2, \dots, b_{92}]$

The accuracy results of Cod1-Cod5 in *H. sapiens* and *M. musculus* datasets were shown in the table below.

Encoding method		Cod1	Cod2	Cod3	Cod4	Cod6
Species						
Homo sapiens constructed dataset		79.95%	88.89%	90.82%	92.51%	89.37%
Mus musculus constructed dataset		86.21%	89.66%	90.52%	94.83%	88.79%
Homo sapiens real dataset		70.77%	78.26%	78.27%	80.92%	74.15%
Mus musculus real dataset		75.00%	69.83%	78.45%	80.17%	66.38%

Note: The values in the table are the accuracy on the independent test set.

Reference:

[1] Yang L, Xia JF, Gui J. Prediction of protein-protein interactions from protein sequence using local descriptors. *Protein Pept Lett.* 2010 Sep;17(9):1085-90.

[2] Shen J, Zhang J, Luo X, Zhu W, Yu K, Chen K, Li Y, Jiang H. Predicting protein-protein interactions based only on sequences information. *Proc Natl Acad Sci U S A.* 2007 Mar 13;104(11):4337-41.

[3] Long Z, Yu G, D Xia, et al. Protein-Protein Interactions Prediction based on Ensemble Deep Neural Networks[J]. *Neurocomputing*, 2018, 324: S0925231218306337-.

- When constructing features of PPI, the paper uses Z-score of the features, but the Z-score uses 1-th eigenvalue of the protein pair, why 1-th eigenvalue is used? Where the eigenvalue is from? is it from the eig-decomposition of the covariance matrix of the feature? Please justify the formula definition.

Thank you for the suggestion. The word "eigenvalue" we express in the paper was not correct, and what we actually want to express is "feature value". So l-th feature value refers to the value of l-th feature.

Z-score method to standardize the feature vector comes from "Advanced Engineering Mathematics". This method unifies data of different levels into the same level for comparison to ensure comparability of data.

- Two ML methods SVM and RF are used in the paper, but I am not sure what is the purpose of SVM in the method. Does the paper combines the two methods by selecting the best one? or just try to explore which ML method is better?

(1) After five-fold cross-validation and independent test set prediction, we know that the prediction performance of Support Vector Machine (SVM) and Random Forest (RF) is very close on the five constructed datasets, including *H. sapiens* constructed dataset, *M. musculus* constructed dataset, *S. cerevisiae* constructed dataset, *D. melanogaster* constructed dataset and *H. pylori* constructed dataset. While the prediction performance of RF is slightly better than SVM on the two real datasets, including *H. sapiens* real dataset and *M. musculus* real dataset.

(2) At the same time, we combined the clear features obtained based on natural vector with other traditional machine learning classification algorithms for prediction and comparison. Different classifier algorithms include SVM, RF, Decision Tree Classifier, KNN, Quadratic Discriminant Analysis, etc. Among them, SVM and RF are better than other classifiers.

- The datasets for negative PPI are from authors' collection and public curated dataset (Negatome), are the manually collected PPI dataset validated by literature searching (or experiments)? If not, the manually collected PPI dataset have more noise. The quality of dataset is very important for ML models.

The negative samples were all experimentally supported non-interacting protein pairs.

REVIEWERS' COMMENTS:

Reviewer #1 (Remarks to the Author):

The authors had addressed all my concerns.

Reviewer #2 (Remarks to the Author):

It is acceptable.

Reviewer #3 (Remarks to the Author):

The revision of the paper has carefully addressed my previous concerns and answered the questions. This paper provides a novel machine learning method for the hard biological problem, especially for prediction non-interactions. I have the following minor comments, and recommend for a minor revision.

- As the revision notes, the gene sequences may contain more information than protein sequences in protein-protein interactions. This is very important features, the research of Gong's on DNA sequence impact of the protein functions and interactions can be cited.
- Please clarify and discuss why the order of two protein sequences in the constructed features has a minor impact on the model accuracy, especially in SVM model.
- The citation of research shall start from the author from the order of the paper. For example, in the revision, 'Stephen Yau, et al' does not match the reference.
- Suggest including one of the major protein-protein approaches, co-evolution, in the introduction.

Dear Editor and Reviewers,

Thank you for your letter and for the reviewers' comments on our manuscript entitled "Protein-protein interaction and non-interaction predictions using gene sequence natural vector" (ID: COMMSBIO-22-0358B). These comments are very helpful for revising and improving our paper. We have carefully studied the comments and corrected them. The responses to the reviewer's comments are as following:

REVIEWERS' COMMENTS:

Reviewer #1 (Remarks to the Author):

The authors had addressed all my concerns.

Reviewer #2 (Remarks to the Author):

It is acceptable.

Reviewer #3 (Remarks to the Author):

The revision of the paper has carefully addressed my previous concerns and answered the questions. This paper provides a novel machine learning method for the hard biological problem, especially for prediction non-interactions. I have the following minor comments, and recommend for a minor revision.

- As the revision notes, the gene sequences may contain more information than protein sequences in protein-protein interactions. This is very important features, the research of Gong's on DNA sequence impact of the protein functions and interactions can be cited.

Thank you for the suggestion. We have cited the research of Gong's on DNA sequence impact on the protein functions and interactions in the Introduction part. (Pages 5 and 6 of the paper)

- Please clarify and discuss why the order of two protein sequences in the constructed features has a minor impact on the model accuracy, especially in SVM model.

Thank you for the suggestion. We have discussed why the order of two protein sequences in the constructed features has a minor impact on the model accuracy, especially in the SVM model in the Discussion part. In addition, we have also compared the prediction performance based on five feature combination methods on *H. sapiens* and *M. musculus* datasets in the Results part. (Pages 13 and 18 of the paper)

- The citation of research shall start from the author from the order of the paper. For example, in the revision, 'Stephen Yau, et al' does not match the reference.

Thank you for the suggestion. We have changed the citation order of the paper and used the first author when citing. (Pages 7, 21 and 22 of the paper)

- Suggest including one of the major protein-protein approaches, co-evolution, in the introduction.

Thank you for the suggestion. We have added the citation of the co-evolution paper ^[1] in the Introduction part. (Page 4 of the paper)

Reference:

[1] Yin, C. & Yau, S. S. A coevolution analysis for identifying protein-protein interactions by Fourier transform. PLoS one 12, e0174862, doi:10.1371/journal.pone.0174862 (2017).

Thank you very much for your comments and suggestions.

Best regards.